# The Multiple Potential Biomarkers for Predicting Immunotherapy Response—Finding the Needle in the Haystack

**DOI:** 10.3390/cancers13020277

**Published:** 2021-01-13

**Authors:** Tamiem Adam, Therese M. Becker, Wei Chua, Victoria Bray, Tara L. Roberts

**Affiliations:** 1Ingham Institute for Applied Medical Research, 1 Campbell St, Liverpool, NSW 2170, Australia; therese.becker@inghaminstitute.org.au (T.M.B.); wei.chua@health.nsw.gov.au (W.C.); 2School of Medicine, Western Sydney University, Campbelltown, NSW 2170, Australia; 3Liverpool Cancer Therapy Centre, Corner of Goulburn and Elizabeth Streets, Liverpool, NSW 2170, Australia; Victoria.bray@health.nsw.gov.au; 4University of New South Wales, Sydney, NSW 2170, Australia

**Keywords:** biomarker, immunotherapy, predictive, NSCLC, urothelial cancer, renal cancer, melanoma

## Abstract

**Simple Summary:**

There have been significant advances in the treatment of cancer within the past 10 years. Immunotherapy is a new type of treatment which uses the body’s own immune system to fight cancer. In some cancers, including lung, melanoma, bladder and kidney, immunotherapy has shown the potential to make people with advanced/metastatic disease live longer. However, most people do not derive any benefit from immunotherapy treatment, and the treatment may cause potentially serious side effects. Currently, our ability to correctly choose who should start these treatments and who is likely to benefit is limited because we do not have many tests to help us make this decision. A number of tests (also known as biomarkers) from tumour tissue, blood and the microbiome have shown promising results. These will be discussed in this review article.

**Abstract:**

Immune checkpoint inhibitors (ICIs) are being increasingly utilised in a variety of advanced malignancies. Despite promising outcomes in certain patients, the majority will not derive benefit and are at risk of potentially serious immune-related adverse events (irAEs). The development of predictive biomarkers is therefore critical to personalise treatments and improve outcomes. A number of biomarkers have shown promising results, including from tumour (programmed cell death ligand 1 (PD-L1), tumour mutational burden (TMB), stimulator of interferon genes (STING) and apoptosis-associated speck-like protein containing a CARD (ASC)), from blood (peripheral blood mononuclear cells (PBMCs), circulating tumour DNA (ctDNA), exosomes, cytokines and metal chelators) and finally the microbiome.

## 1. Introduction

In the last 10 years, immune checkpoint inhibitors (ICIs) have emerged as a new systemic therapy option for a number of advanced cancers, with up to 50% of patients deriving benefit, although in many cancers this percentage is much lower [1,2]. This means that a significant proportion of patients may not derive any benefit from these treatments. This is a concern since potentially serious immune-related adverse events (irAEs) can occur in up to 15–25% of patients [1,3,4]. Additionally, these treatments can be prohibitively expensive and with increasing indications for its use, this will place a significant strain on healthcare systems. 

The number of biomarkers available to predict response to ICIs is very limited, with tumoural programmed cell death ligand 1 (PD-L1) and tumour mutational burden (TMB) the only biomarkers currently used in clinical practice [1,5]. However, the reliance on tumour tissue and the heterogeneous nature of these biomarkers limit their use. There is, therefore, a critical need to develop less invasive biomarkers that could be used to personalise ICI therapy in patients with advanced cancers. 

Here, we review and discuss the current landscape of ICI therapy with a particular focus on non-small cell lung cancer (NSCLC), melanoma, renal cell carcinoma (RCC) and urothelial/bladder carcinoma. We will also discuss the current state of biomarkers in the field of ICI therapy. 

## 2. Overview of ICIs

### 2.1. History of ICIs 

Immune checkpoints are a diverse group of proteins whose function is to restrict physiologic immune cell responses in order to maintain immune homeostasis and prevent normal host tissues from developing collateral damage due to inflammation [6]. Examples of immune checkpoint proteins include cytotoxic T-lymphocyte associated protein 4 (CTLA-4) and programmed death 1 (PD-1) (and its ligand programmed death ligand 1/2 (PD-L1/PD-L2)).

In 1996 Dr James Allison’s team showed that blockade of CTLA-4 could lead to enhanced anti-tumour immune response and tumour rejection. Two years later in 1998–1999 Dr Tasuku Honjo’s team showed that PD-1 is another regulator of immune response by knocking out the PD-1 gene in mice; their work in this field culminated in them sharing the 2018 Nobel Prize in Physiology or Medicine [7]. 

CTLA-4 is a surface receptor on T lymphocytes which serves to suppress T cell activation and expansion on cluster of differentiation CD4+ and CD8+ T cells by binding to co-activator proteins CD80/CD86 in greater affinity than CD28 on antigen-presenting cells (APCs) [8]. 

PD-1, expressed on T cells, and its ligand PD-L1, expressed mainly on tumour cells, are designed to inhibit T cell response [8]. Binding of PD-1 to its ligand PD-L1/PD-L2 leads to T cell death [8]. T cell exhaustion is a state where chronic stimulation through this pathway can lead to dysfunctional cells unable to respond further with reduced ability to proliferate, release cytokines or exert their cytotoxic function [8]. 

Apart from the CTLA-4 and PD-1/PD-L1 axis, other immune checkpoints and co-signalling molecules exist, including: lymphocyte activation gene 3 (LAG-3), T cell immunoglobulin mucin 3 (TIM-3), indoleamine 2,3-dioxygenase 1 (IDO-1), B and T lymphocyte attenuator (BTLA) member of the immunoglobulin superfamily, V-domain immunoglobulin suppressor of T cell activation (VISTA), T cell immunoreceptor with Ig and ITIM domains (TIGIT), adenosine A2A receptor (A2AR), 4–1-BB, OX-40 and CD27 members of the tumour necrosis factor receptor family, CD40 and inducible T cell co-stimulator (ICOS) [9]. 

The first ICI approved by the Food and Drug Administration (FDA) was the anti-CTLA-4 antibody ipilimumab for unresectable or metastatic melanoma in 2011. Since that time, a number of other agents have been approved (Table 1). 

### 2.2. irAEs

The Common Terminology Criteria for Adverse Events (CTCAE) Version 5 are a set of criteria used to classify adverse effects of anticancer therapies. It uses a range of grades to specify the severity of the adverse effect (1 = mild, 2 = moderate, 3 = severe, 4 = life threatening and 5 = death).

ICIs can cause irAEs, with up to 15–25% of patients experiencing potentially severe side effects. Although lower grade toxicities (Grade 1–2) can usually be managed by temporary discontinuation of ICI therapy, more severe toxicities (Grade 3–4) usually require systemic steroids and, in some instances, permanent discontinuation of treatment. The most common irAEs are summarised in the Table 2 below [10]: 

Less common irAEs: other endocrinopathies (hypophysis, primary adrenal insufficiency, hypogonadism, pancreatitis, hypercalcaemia, type 1 diabetes mellitus), pneumonitis (more common in combination therapy and if pre-existing lung disease or thoracic radiotherapy), arthritis (arthralgias, rheumatoid-like polyarthritis and reactive arthritis), Sicca syndrome (severe dry mouth), nephritis, myositis, myocarditis, sarcoidosis, coeliac disease, polymyalgia rheumatica, giant cell arteritis, myasthenia gravis, transverse myelitis, uveitis, episcleritis, pericarditis and Takotsubo-like cardiomyopathy [10].

## 3. ICI Therapy in NSCLC, Melanoma, RCC and Urothelial/Bladder Cancer

### 3.1. NSCLC

Lung cancer accounts for 11.6% of cancer diagnoses and is the leading cause of cancer-related deaths worldwide [11]. In spite of all the recent advances in cancer treatment, 5-year overall survival (OS) remains poor at <20% mainly due to advanced stage of disease at diagnosis [12]. Lung cancer can be subdivided into small cell lung cancer (SCLC) (15%) and NSCLC (85%). Of all NSCLC diagnoses, adenocarcinoma accounts for 40–50% while squamous cell carcinoma (SCC) accounts for 20–30% [13]. Prior to ICIs, chemotherapy was the standard of care treatment for patients with non-mutant NSCLC. For patients with epidermal growth factor receptor gene (EGFR) mutation, anaplastic lymphoma kinase (ALK)/echinoderm microtubule like 4 (EML4) fusion and ROS1 fusion, targeted therapy using tyrosine kinase inhibitors (TKIs) is the standard of care treatment. 

For patients with metastatic (stage IV) NSCLC, pembrolizumab was compared with standard of care platinum doublet chemotherapy in the first line setting (Keynote-024) [1]. Results of this study showed that in patients with tumoural PD-L1 expression >50%, pembrolizumab was superior to chemotherapy in terms of overall response rate (ORR), progression-free survival (PFS) and OS. In the second line setting nivolumab (Checkmate-017, Checkmate-057), pembrolizumab (Keynote-010) and atezolizumab (POPLAR) have all shown an ORR, PFS and OS benefit compared with docetaxel chemotherapy [14]. Importantly, this benefit was seen independent of PD-L1 expression screened in matching tumour tissue. As a result, ICIs have become the standard of care for patients with non-mutant NSCLC who have progressed on first line platinum doublet chemotherapy and who have not received prior ICIs. 

In the setting of locally advanced (Stage II–III) NSCLC patients who have had definitive (curative intent) chemoradiotherapy and who did not progress within 12 months, the PACIFIC trial showed that 12 months of durvalumab was superior to placebo regardless of PD-L1 expression [14]. 

There is increasing evidence of the synergistic activity of chemotherapy when combined with ICIs. This could partly be explained by chemotherapy’s immunological effects such as reducing regulatory T cell (Treg) activity, reducing myeloid-derived suppressor cells (MDSCs) and enhancing cross-presentation of tumour antigens [15]. Two recent publications (Keynote-189 and Keynote-407) showed improved PFS and OS when combining ICIs with chemotherapy versus chemotherapy alone. In both studies, response was independent of PD-L1 expression [16]. Another publication (IMPOWER 150) showed similar results with the combination of ICIs, chemotherapy and bevacizumab [17]. 

### 3.2. Melanoma

Over the last 10 years, the annual cases of melanoma have increased by nearly 50% to over 287,000 worldwide, with more than 60,000 melanoma-related deaths per year [11]. There are four major subtypes of invasive cutaneous melanoma: superficial spreading is the most common, accounting for 70%; nodular accounts for 15–20%. Other subtypes include lentigo maligna and acral lentiginous. 

Surgical excision remains the treatment of choice for early cutaneous melanoma and is curative in most cases. However high-risk features in the primary melanoma and/or spread to regional lymph nodes define a subset of patients at high risk of disease recurrence and disseminated spread. 

For most patients with resected stage III melanoma, adjuvant ICI therapy with nivolumab or pembrolizumab has been shown to improve recurrence-free survival and is recommended [18,19,20]. While for resected stage IV disease, the combination of nivolumab and ipilimumab is the preferred option [21]. In unresectable/stage IV melanoma, ICI therapy is the preferred option with either single agent pembrolizumab or nivolumab, while the combination of nivolumab and ipilimumab still has a role but is associated with up to 40–50% risk of Grade 3 toxicity [22,23,24]. Expression of PD-L1 was assessed in the studies evaluating ICI therapy in melanoma but was not found to be useful as patients derived benefit regardless of PD-L1 expression. Treatment should be personalised based on individual cases. 

### 3.3. RCC 

RCC accounts for 2–3% of all cancers and up to 338,000 new cases each year worldwide [25]. Around 30% will present with metastatic disease at diagnosis, and approximately one third of those who initially have curative treatment progress to advanced disease [25]. Clear cell histology accounts for approximately 75% of RCC followed by papillary (20%) and chromophobe (5%) [25]. 

Checkmate 214 (a phase III trial) compared the combination of ipilimumab and nivolumab versus standard of care sunitinib in patients with metastatic RCC. In the intermediate/poor risk group of patients (according to the International Metastatic RCC Database Consortium (IMDC) prognostic score), ICI therapy had improved ORR, OS and led to less Grade 3–4 toxicity. PD-L1 expression was evaluated as an exploratory outcome, although ORR and PFS were longer in those with PD-L1 >1%; even patients with PD-L1 <1% benefited from ICI therapy compared with sunitinib and OS was longer across all PD-L1 expression levels for ICI therapy. In the second line setting, nivolumab was compared with standard of care everolimus (Checkmate-025); nivolumab was found to be superior in terms of PFS and OS. Additionally, nivolumab had a better toxicity profile, with a smaller number of Grade 3 or higher adverse events [25]. PD-L1 expression was not evaluated in this study. 

### 3.4. Urothelial/Bladder Cancer 

Bladder cancer is the ninth most common cancer diagnosed worldwide affecting 430,000 and causing 165,000 deaths each year [26]. The greatest risk factor remains smoking [26]; other risk factors include age, male sex, Caucasian ethnicity and workplace exposure to industrial chemicals (benzidine, beta-naphthylamine, chemicals in rubber and textile industries) [27]. Urothelial carcinoma (UC) of the bladder accounts for 90% of all urothelial cancers with the remaining 10% in the ureter, urethra and urachus [28]. The majority (around 75%) of bladder cancers are non-muscle invasive (NMIBC); although up to half of those treated with transurethral resection have recurrence with up to one quarter progressing to muscle invasive disease [28]. 

Although the FDA has approved five ICIs in the second line setting, only pembrolizumab has been validated in a randomised controlled trial (Keynote-045). The remaining four agents (nivolumab, atezolizumab, durvalumab and avelumab) were approved based on single arm studies versus historical records [29]. The recently published JAVELIN Bladder 100 trial showed that the addition of avelumab as maintenance therapy following completion of first line standard of care chemotherapy had significantly improved OS irrespective of PD-L1 status [30].

Table 3 summarises ICIs used in NSCLC, melanoma, RCC and urothelial bladder cancer as well as the indication and supporting evidence. 

## 4. Tumour-Derived Biomarkers

### 4.1. PD-L1

PD-L1 expression is regulated by constitutive (intrinsic) and induced (extrinsic) pathways. Other mutations (e.g., Janus kinase (JAK3)) activating mutations can also increase the expression of PD-L1 [31]. 

PD-L1 expression is most commonly assessed using immunohistochemistry (IHC) on tumour tissue. There are currently five different anti-PD-L1 antibody clones available for diagnostic IHC: 22–8 (Dako), 22C3 (Dako), SP142 (Ventana), SP263 (Ventana) and 73–10 (Dako). Depending on which clone is used, PD-L1 cut-offs used and detection rates vary slightly [31]. Blood-based “soluble” PD-L1 (sPD-L1) has been investigated by different research groups and will be discussed later on in the blood-derived biomarkers chapter. 

A number of studies have extensively looked at PD-L1 expression as a predictive biomarker for ICI therapy with conflicting results. A phase 1 trial of 42 patients with advanced cancers (melanoma, NSCLC, colorectal, RCC and prostate cancers) treated with ICIs showed a response rate of 36% for patients positive for PD-L1 expression vs. 0% if PD-L1 expression was negative. Keynote-001 showed that in patients treated with pembrolizumab, increased PD-L1 expression was associated with better response and PFS [32,33]. 

Given that PD-L1 expression is a mechanism of immune system evasion that is utilised by tumour cells, overexpression of PD-L1 can therefore be postulated to correlate with worse prognosis across different tumours [6]. Limitations of PD-L1 as a predictive biomarker include the spatial and temporal heterogeneity of PD-L1 expression within the tumour [32], as well as different assays, each with different cut-offs. Biopsy samples taken months or years prior to eventual commencement of immune checkpoint inhibition may not reflect current expression status, especially if patients have been exposed to chemotherapy, radiotherapy or other anti-cancer treatments [31]. 

Additionally, a number of publications have shown improved response rates and survival outcomes in a number of different cancers treated with ICIs even with negative PD-L1 expression [17,34]. 

### 4.2. TMB

TMB is the total number of non-synonymous somatic mutations of the genomic coding area. Non-synonymous somatic mutations alter the amino acid sequence of proteins encoded by the affected gene, resulting in the formation of neoantigens. It is hypothesised that higher TMB creates a larger number of neoantigens which sets off the cascade of immune cell response including effector T cells leading to stronger response to ICIs. This usually excludes germline mutations which the body recognises as normal self. Pre-clinical data in melanoma and NSCLC cell lines indicate that high-load frameshift indel (insertion and deletion mutations) are highly immunogenic compared with single nucleotide variant load [31]. This is an interesting finding which suggests that not only the quantity of mutations but also the mutation type is important. 

TMB can be affected by a number of factors, including hereditary (microsatellite instability (MSI) due to deficient mismatch repair (MMR)), epigenetic changes (i.e., modifications to the DNA of cancer cells that do not involve a change in the nucleotide sequence, e.g., DNA methylation and histone modifications) and environmental (smoking, diet, UV light and exposure to other chemicals). 

TMB is most commonly analysed using next generation sequencing (NGS) panels; whole exome sequencing (WES) is another option, although this technique is not widely available and is costly and time-consuming [14]. 

In a number of publications, TMB was found to be higher in cancers with high mutagen exposure such as NSCLC (average 7.2 mut/Mb) and melanoma (average 13.5 mut/Mb) and lower when a driver mutation is present [31]. A study by Rizvi et al. showed that higher mutational burden in NSCLC patients who received pembrolizumab led to a better ORR and greater durable clinical response (>6 months). In patients with advanced urothelial cancer who received atezolizumab, median mutational load was significantly higher in the responders (12.4 Mut/Mb) vs. the non-responders (6.4 Mut/Mb) [35]. Goodman et al. showed that the quality of the mutational burden (i.e., clonal vs. sub-clonal) also had an impact on outcomes, with clonal mutations (homogenous tumours) being associated with better outcomes compared with sub-clonal (heterogeneous tumours) [8]. An important factor to note is that different studies used different cut-offs for “high TMB”; although generally >13 mut/Mb was considered high in most studies. Blood-based TMB analysis using ctDNA and other methods is becoming an attractive option given the “less-invasive” nature and ability of repeat sampling during treatment [31,36]. 

### 4.3. STING

The STING pathway surveys for the presence of cytosolic DNA; cyclic-GMP-AMP (cGAMP) synthase (cGAS) senses cytosolic DNA and catalyses the synthesis of cGAMP. cGAMP, the endogenous ligand of STING then induces conformational changes in STING which causes its subsequent trafficking from endoplasmic reticulum to perinuclear golgi. This results in recruitment and phosphorylation of TANK-binding kinase 1 (TBK1) which in turn phosphorylates and activates interferon regulatory factor 3 (IRF3) and nuclear factor kappa-light-chain-enhancer of activated B cells (NF-κB), leading to activation of type 1 interferon (IFN) transcription [37,38,39]. Subsequently, type 1 IFN leads to T cell differentiation towards the Th1 response, activation of effector CD8+ T cells, as well as activation and maturation of dendritic cells [40]. 

There are multiple mechanisms the body uses to regulate the STING pathway, including elimination of accumulated DNA by DNases and post-translational modification of proteins [37]. Recent research also suggests that through dual stimulation of two innate immune pathways that have opposing functional activity, cytosolic DNA can activate STING as well as the absent in melanoma 2 (AIM2) inflammasome in antigen presenting cells (APCs) [37]. AIM2 senses DNA and forms the inflammasome complex with the adaptor molecule ASC, leading to caspase-1 activation and eventually IL-1β production and pyroptosis (inflammatory cell death) [41,42]. 

The STING pathway is also activated following exposure to DNA damaging agents (such as chemotherapy or radiotherapy) [43]; this can occur either via direct activation from the site of DNA damage or via the accumulation of cytosolic DNA resulting from substantial unrepaired DNA damage [44,45,46]. The STING pathway appears to be an important innate sensing pathway for detection of tumours [43]. STING pathway activation within APCs in the tumour microenvironment (TME) drives T-cell priming against tumour associated antigens [47]. Activation of this pathway also upregulates PD-L1 expression on cancer cells and tumour infiltrating lymphocytes (TILs) following DNA damage, thereby dampening immune mediated tumour killing. This provides a potential biological rationale for combining a STING agonist and/or DNA damaging agents with ICI therapy [43]. 

However, STING signalling is inhibited in a number of different cancers through hypermethylation processes [48], while STING polymorphisms or alterations of regulatory proteins may affect its function. In mouse models, STING deficient mice had weaker response to topotecan chemotherapy [49]. In another study, STING deficient mice had reduced anti-cancer radiotherapy efficacy [49]. Most relevant for this review, studies have demonstrated that the therapeutic effect of CTLA-4 and PD-L1 inhibitors was lost in STING deficient mice [49]. 

Other work showed that STK11/LKB1 inactivation could be a major genetic driver of primary resistance to immune checkpoint inhibitor therapy. STING transcriptional silencing via mutational STK11/LKB1 inactivation led to insensitivity to cytoplasmic double stranded deoxyribonucleic acid (dsDNA) detection and an inability to mount an appropriate immune response [50]. In addition, this mutation in non-squamous NSCLC cell lines predicted worse outcomes after treatment with immune checkpoint inhibitor therapy [51]. 

The STING pathway is a cascade of events that eventually leads to the activation of Batf3-lineage dendritic cells (DCs) which appear to be central to anti-tumoural immunity [37]. STING expression might prove to be a useful predictor of response to ICIs. It may be possible to manipulate this pathway to achieve better results; for example, injection of Batf3 dendritic cells, STING agonists (many are in development) [52] and radiotherapy (DNA damage activates STING pathway) [53]. 

### 4.4. ASC

ASC is an adaptor molecule which along with a protein sensor and the protease pro-caspase-1 form the inflammasome complex [54]. These molecules form a complex when activated (by pathogen-associated molecular patterns (PAMPs), damage-associated patterns (DAMPs) and other cytosolic disturbances) and quickly accumulate to form a large cytosolic oligomer; this allows cleavage of pro-caspase-1 to the active form of caspase-1; caspase-1 then cleaves pro-IL-1β and pro-IL-18 to their mature forms and induces pyroptosis (lytic inflammatory cell death) by activation of gasdermin D (canonical pathway). Alternatively, pro-caspase-11 can detect cytosolic lipopolysaccharide (LPS) from gram-negative bacteria; this leads to cleavage of pro-caspase-11 to the active form of caspase-11 which induces gasdermin D cleavage and consequent pyroptosis (non-canonical pathway) [49,55]. Inflammasomes can be found in many different cells, including dendritic cells, macrophages as well as glial, endothelial and neuronal cell lineages [49]. 

ASC is a tumour suppressor gene which is frequently inactivated by de novo promoter methylation in up to 40% of NSCLC patients [56]. Patients without ASC promoter methylation showed significantly better survival rates [56]. In renal cancer cell lines, up to 41% of tumour cells had ASC gene methylation compared with only 12% of adjacent normal tissue. Methylation significantly correlated with higher tumour grade, while restoration of the ASC gene through demethylation treatment improved sensitisation to DNA damaging agents [57]. 

In melanoma mouse models, ASC appears to have stage-dependant dual roles in tumourigenesis [58]. In primary melanoma, ASC expression inhibits tumour growth by reducing IKKα/β phosphorylation and inhibiting NF-κB activity. Whilst in metastatic melanoma, ASC promotes tumour growth through enhanced NF-κB activity and inflammasome mediated IL-1β secretion [58]. Another interesting finding in mice deficient in inflammasome components (ASC, caspase-1, AIM2) is that they can develop an altered intestinal microbiota [59]. Another study showed that mice deficient in the DNA sensor AIM2 were protected from the effects of radiation since AIM2 mediates the caspase-1-dependant death of intestinal epithelial cells and bone marrow cells in response to double-stranded DNA breaks caused by ionising radiation and chemotherapy [59]. 

### 4.5. TME

The TME is a complex system that includes immune cells, stromal cells, tumour cells as well as chemokines, cytokines and other signalling molecules. The composition of the TME and interplay between different cells and molecules has an impact on tumour growth as well as response to ICIs. The immune contexture refers to the functional orientation, location and density of the immune cell infiltrate and can be prognostic of patient outcomes in different cancers [31]. 

Another categorisation that can be useful when considering the TME includes these three categories; immune inflamed (hot tumour, with high immune cell infiltrate in the tumour core and invasive margin), immune desert (cold tumour, devoid of immune infiltrate) or immune excluded (immune infiltrate resigned to the stroma unable to enter the tumour core or invasive margin) [60]. Inflamed tumours have been correlated with better response rate to ICIs and improved survival, whilst immune desert and immune excluded tumours have been associated with resistance to ICIs and worse outcomes [60]. Important cytokines seen in the inflamed cancer phenotype capable of recruiting effector T cells include CCL2, CCL3, CCL4, CCL5, CXCL9 and CXCL10. CXCL9 and CXCL10 are most likely responsible for T cell recruitment as they are recognised by CXCR3 on effector CD8+ T cells [37]. 

TILs play an important role in a number of different cancers and in terms of response to ICIs. In particular, higher densities of T cells (CD3+), cytotoxic T cells (CD8+) and memory T cells (CD45RO+) have been correlated with better survival [31,61]. Conversely, higher densities of Foxp3 regulatory T cells are associated with worse survival and increased recurrence [31]. In a study of 65 advanced NSCLC patients, baseline levels of PD-L1, CD3+ T cells, CD4+ T cells and CD8+ T cells were associated with improved response rates indicating the potential predictive value of TILs in ICI therapy [31]. In another study of metastatic melanoma patients treated with ICIs, an increase in CD8+ T cells from baseline to post-treatment biopsy (specifically at the tumour core and invasive margin) was significantly associated with radiological reduction in tumour size [6]. In particular, the CD4+ regulatory T cells (Tregs) expressing the transcription factor Foxp3 are highly immune suppressive and promote tumour progression by suppressing effective anti-tumour immunity [62]. 

Tumour associated macrophages (TAMs) have been associated with poor PFS and OS in a number of studies looking at patients treated with ICIs [63]. When exposed to a range of pathogenic and environmental signals, macrophages can take on two major phenotypes. M1 macrophages can be generated by microbial products and interferon gamma (IFNγ) (and are thereby associated with pro-inflammatory cytokines and enhance the immune response). M2 macrophages develop in response to IL-4, IL-13 and IL-10. In the TME they contribute to cancer progression, angiogenesis, metastases and immunosuppression [64,65]. 

## 5. Blood-Derived Biomarkers—“Liquid Biopsies” 

### 5.1. Introduction

Peripheral blood-derived biomarkers serve as an attractive option given the “less-invasive” nature of their collection compared with tissue and the ability to repeat them whilst patients are on treatment. 

### 5.2. PBMCs and Other Blood Cells 

A PBMC is any peripheral blood cell that has a round nucleus. These cells can consist of lymphocytes (T cells, B cells, NK cells) and monocytes. Erythrocytes and platelets have no nuclei whilst neutrophils, basophils and eosinophils have multi-lobed nuclei. In humans, lymphocytes make up the majority of the PBMC population followed by monocytes and finally a small percentage of dendritic cells. Different methods can be used to evaluate PBMCs such as flow cytometry and mass cytometry. 

PBMC immune profiling using flow cytometry of patients with melanoma treated with ICIs showed that patients with higher number of CD4+ and CD8+ T cells had better response than those with lower numbers, and that perhaps the ratio of T effector cells to T regulatory cells may be a good predictor of response [66,67]. Additionally, results from peripheral blood analysis correlated with tissue levels of CD8+ T cells [67]. A newer technique known as mass cytometry can simultaneously detect up to 40 markers to identify different cell subsets and determine their function by expression of cytokines, cytotoxicity and activation markers [68]. In a patient with metastatic melanoma who developed severe hepatitis due to ICI therapy, mass cytometry was able to identify a subset of CD4+ T cells expressing abnormally high multi-drug resistance type 1 transporter (MDR1) activity which has been implicated in steroid resistance [69]. 

In addition to PBMCs, other blood cells have shown promise as potential predictive and prognostic biomarkers. Different absolute values and ratios have been studied on peripheral blood including Absolute Neutrophil Count (ANC), Absolute Lymphocyte Count (ALC) and Neutrophil to Lymphocyte Ratio (NLR). In one study of 134 advanced NSCLC patients treated with nivolumab, an ALC of ≥1000/µL was associated with significantly improved OS and PFS while an ANC of ≥7500/µL was associated with worse OS and PFS [6,31]. In another study of 175 advanced NSCLC patients treated with nivolumab, an NLR of ≥5 was associated with worse OS and PFS [31,70,71,72,73,74]. A meta-analysis of 15 cohorts and >3000 RCC patients showed that a high NLR was associated with poor prognosis [75]. 

Other potential biomarkers that have been studied include baseline eosinophil count and lactate dehydrogenase (LDH). High baseline eosinophil count and low LDH count were associated with improved survival in melanoma patients treated with pembrolizumab [6,8]. LDH is the final enzyme in the glycolysis pathway and catalyses the conversion of pyruvate to lactate. High levels of LDH have been reported in cancer patients due to the Warburg effect (increased utilisation of glycolysis rather than oxidative phosphorylation for tumour energy requirement). In a study of 230 metastatic melanoma patients treated with Ipilimumab, higher LDH was seen in patients who had worse median OS [32]. C-reactive protein (CRP) is a non-specific indicator of acute inflammation and has been correlated with poor prognosis, likely due to the presence of neutrophils and tumour suppressive cytokines. In a study of 1161 RCC patients who underwent nephrectomy, CRP using multivariate analysis was seen as an independent prognostic factor of cancer-specific survival [32]. 

Myeloid-derived suppressor cells (MDSCs) are a heterogeneous group of immune cells from the myeloid lineage (originating from bone marrow stem cells). These cells have been implicated in resistance to ICI therapy and worse survival outcomes. Reduced numbers of MDSCs in blood were associated with improved PFS [8]. It is thought that they inhibit T cell activation through expression of nitric oxide synthase 2 (NOS2) and arginase (ARG1). They also induce development of Tregs and polarisation of macrophages to TAM-like phenotype [76]. 

### 5.3. ctDNA

Cell-free DNA (cfDNA) is derived from dying cells (in individuals without cancer, most cfDNA is from haematopoietic cells); this is different from ctDNA which originates from tumour cells [77]. ctDNA may include genetic information from the entire tumour genome (i.e., including multiple subclones and metastatic sites) and therefore provides insight into clonal heterogeneity and evolution of all solid cancers present at any one time [78]. 

ctDNA has a short half-life of approximately 2 h and usually comprises fewer than 200 nucleotides. Multiple methods are available for detection, including qPCR, BEAMing, droplet digital PCR (ddPCR) and NGS [7]. 

Patients with metastatic melanoma treated with ICIs had ctDNA screened for using ddPCR at baseline and were then re-evaluated at the 8 and 12-week mark. Interestingly, reduction in ctDNA levels at 8 and 12 weeks predicted response to ICIs, radiological progression and ability to differentiate between pseudoprogression and real disease progression [79,80]. In another study, 45 patients with advanced NSCLC treated with nivolumab had ctDNA analysed at baseline and after 6 weeks an increase in ctDNA >20% at the 6-week mark was significantly associated with worse OS and median time to progression [81]. In another study of patients with advanced melanoma treated with ICIs, 22 patients had ctDNA detected using ddPCR, tumour progression was detected with 100% specificity and, in 13 out of 16 patients early increase in ctDNA levels was associated with shorter PFS [82]. 

Another area where ctDNA has shown promise as a biomarker is differentiating pseudoprogression from true progression in patients receiving ICIs. Pseudoprogression is defined as an increase in the size of the primary tumour or the appearance of a new lesion followed by tumour regression [83]; this is not true progression and is often due to an immune cell infiltrate. An explorative biomarker study in 125 metastatic melanoma patients receiving ICIs demonstrated that ctDNA profiles can accurately differentiate pseudoprogression from true progression [84]. 

### 5.4. Exosomes

Exosomes are extracellular vesicles that can be released by cells including both tumour and immune cells; they are membranous vesicles derived from within the endosomal compartment of a cell [7] and are capable of carrying significant amounts of information in the form of mRNA, miRNA and proteins between cells [61]. Their cargo is influenced by the origin cell and can carry both immunostimulatory and immunosuppressive signals [6,61]. Exosomes may also transfer unique patterns of their contents to neighbouring cells and may thereby induce the phenotypic modifications in the recipient cells, subsequently modulating the microenvironment [85]. Different methods are available to isolate exosomes: size exclusion chromatography (SEC), differential centrifugation and sucrose gradient centrifugation (SGC), and more recently commercial kits are available for isolation [85,86]. The cargo of isolated exosomes can then be examined by various procedures, including mass spectrometry, Western blots and flow cytometry [87]. 

Exosomes have been implicated in a number of different cancers (ovarian, prostate, breast); they are specifically involved in development of metastases through their effects on the Epithelial to Mesenchymal Transition (EMT) aiding in angiogenesis, extracellular matrix degradation, immune system suppression and even drug resistance [7,86]. 

Early results from a trial of patients with metastatic melanoma treated with ICIs suggested that PD-L1 expression on circulating exosomes can change whilst on treatment, indicating response to therapy [6,31]. In another study of 18 head and neck cancer patients treated with cetuximab, ipilimumab and radiotherapy, exosome molecular cargo (isolated from plasma by SEC) was increased in those that had disease recurrence and decreased in those that remained disease-free, indicating possible predictive value [88]. In a further study of 59 patients with metastatic melanoma treated with ipilimumab, levels of exosomes from both T cells and dendritic cells were obtained; a significant increase in the PD-1 and CD28 expression was seen in those having clinical benefit with improved PFS and OS [89]. 

More recently, studies looking specifically at exosomal PD-L1 have shown it is a marker of poor ICI outcomes in patients with gastric and head and neck cancer who underwent surgery, chemotherapy and radiation therapy [90]. In metastatic melanoma patients, exosomal PD-L1 distinguished patients who responded to PD-1 therapy from those that did not based on an increase in its concentration; however, what wasn’t clear was whether the PD-L1 originated from tumour cells or immune cells [90]. 

Exosomes are extremely stable under various conditions such as freezing, cold storage for up to years and thawing, and are abundant in plasma [85]. This makes them appealing as potentially feasible biomarkers in the future. 

### 5.5. Cytokines

Cytokines are important regulators of the immune system; they are able to recruit immune cells to the TME and induce expression of immune checkpoint receptors such as TIM-3 and PD-1 [91]. A recent study of 98 patients with metastatic melanoma had 65 cytokines profiled as part of a 65-plex discovery assay. Eleven cytokines were found to be significantly upregulated in patients who experienced severe immune-related adverse events; these 11 cytokines (G-CSF, GM-CSF, Fractalkine, FGF-2, IFNα2, IL12p70, IL1a, IL1B, IL1RA, IL2, IL13) were integrated into a single cytokine toxicity score (CYTOX) and validated its ability to predict immune-related adverse events [92]. 

### 5.6. Metal Chelators

A recent publication showed a strong correlation between copper transporter 1 (CTR-1) and PD-L1 expression in a number of different cancers but not in normal tissue [91]. Copper chelating drugs (such as TEPA) led to an increase in CD8+ T cells and NK cells and slowed tumour growth and improved mice survival [91]. Copper chelating drugs inhibit phosphorylation of STAT3 and EGFR and promoted ubiquitin-mediated degradation of PD-L1 [91]. Further work in this field by Vittorio et al. looked at Catechin (a natural polyphenol) which targets copper metabolism and induced oxidative stress in neuroblastoma cell lines. Subsequent depletion of copper appeared to successfully inhibit angiogenesis [93,94]. 

### 5.7. Soluble PD-L1 and PD-1

Numerous costimulatory molecules in immunoregulation pathways assume two forms of expression; the membrane-bound and soluble forms [95]. It is postulated that circulating sPD-L1 arises from intrinsic splicing activities in tumour cells and the proteolytic cleavage of membrane-bound PD-L1 [96]. The most common method to detect sPD-L1 is by using enzyme-linked immunosorbent assay (ELISA), kits available commercially. sPD-L1 was studied in a prospective study of 39 NSCLC patients treated with nivolumab; results showed that 59% of patients with low plasma sPD-L1 at baseline achieved complete or partial response vs. 25% of those with high baseline levels. In addition, 22% of patients with low plasma sPD-L1 levels developed disease progression vs. 75% of those with high plasma sPD-L1 levels. Time to treatment failure (TTF) and OS were significantly longer in those with low plasma sPD-L1 levels compared with high levels [97]. Similarly, sPD-1 was evaluated in 38 NSCLC patients with an EGFR mutation treated with erlotinib. The serum concentration of sPD-1 increased during erlotinib treatment, and an increase in sPD-1 during treatment was associated with prolonged PFS and OS [98]. 

### 5.8. Circulating Tumour Cells (CTCs)

CTCs may reflect disease burden and their presence is usually associated with poor prognosis in a number of different cancers, including breast, prostate, lung and colorectal [99]. They usually present in small numbers and for a short period of time; therefore, their detection can be difficult. Some evidence, however, suggests that their change over time can be predictive of response to ICIs [99,100,101]. 

## 6. The Microbiome 

The microbiome is the collective genomes that can be found within a single microbial ecosystem. The microbiota is the community of microorganisms that inhabit all the surfaces in an organism that are exposed to the external environment. It includes bacteria, fungi, protozoa and viruses [102]. The microbiome contains up to 100 trillion microbes, with 5000 distinct species and accounts for up to 2 kg of the human body weight [7]. Bacteroidetes and Firmicutes constitute 90% of the intestinal microbiome while Proteobacteria and Actinomycetes constitute 10% [36,103]. 

A number of factors can affect normal microbiome composition, including race, ethnicity, pregnancy, hormonal changes, sexual activity and hygiene [7]. Other factors that can affect microbiome development include birth mode (vaginally vs. C-section), feeding mode (breast feeding vs. bottle feeding), antibiotic use in early life, preterm birth, host genetics as well as other environmental factors (lifestyle, geographical location, religion and culture) [7]. 

The gut microbiome can be analysed using different techniques such as 16S rRNA gene sequencing of hypervariable regions. Other techniques include shotgun sequencing of the whole genomic DNA (mapping against metabolic pathway databases can provide further insight into the function of the bacterial communities) [104]. 

Microbial pathogens may drive tumourigenesis in up to 15–20% of cancer cases [79]. The gut microbiome interacts with immune cells and gut cells through activation of innate immune signalling pathways. In the presence of the microbiota, host antigen presenting cells activate interferon (IFNγ) producing T cells which are enriched during ICI treatment. This activation can occur at distant sites. It remains unclear whether microbial metabolites disseminate systemically and reach tumour compartments or whether they act on peripheral lymphoid organs [102]. Vetizou et al. identified several Bacteroides species that were capable of promoting maturation of intra-tumoural dendritic cells and inducing type 1 T helper cell activation in the draining lymph nodes [102]. 

In patients with melanoma who received anti-PD-1 therapy, Faecalibacterium species was enriched in responders with associated increased levels of effector T cells, reduced frequency of peripheral Treg cells and reduced number of MDSCs. Enrichment of Bacteroides species was seen in non-responders [31]. In another cohort of patients with advanced NSCLC who received anti-PD-1 therapy, Akkermansia muciniphila was abundant in those that responded to treatment. A study by Gopalakrishnan et al. of 112 patients with metastatic melanoma treated with anti-PD-1 therapy found that the gut microbiome of responders was more diverse than in non-responders, as well as being associated with improved PFS. Responders were enriched with Clostridiales, Ruminococcaceae and Faecalibacterium while non-responders were enriched with Bacteroidales [6].

The effects of antibiotic use on the microbiome and potential impacts on ICI response were evaluated in two studies of patients with NSCLC, RCC and urothelial cancer (in one of the two studies); they both concluded that patients who received antibiotic treatment within 30 days of commencing ICI therapy were seen to have increased risk of early progression, reduced PFS and OS [105,106]. 

Several methods can be utilised to manipulate the microbiome, including probiotics (live microorganisms that can confer health benefits to the host), prebiotics (substrate(s) that (are) selectively utilised by host microorganisms conferring a health benefit) [106] and faecal microbiota transplantation. In a study of melanoma cancer cells, faecal microbiome transplantation from patients with responding versus non-responding disease into germ-free mice showed a delay in tumour growth and enhanced response to ICIs [104]. 

Section 4, Section 5 and Section 6 have extensively described tumour, blood and microbiome-derived biomarkers. Table 4 (below) summarises the pertinent points with the relevant references. 

## 7. Discussion

Over the past few years, the term ‘immunotherapy’ has become widely adopted in the field of oncology and usually refers more specifically to drugs that inhibit PD-1, PD-L1 and CTLA-4 immune checkpoints. Immunotherapy has become the standard of care in a number of different cancers, including advanced NSCLC and melanoma [1,22,23,24]. With a growing body of evidence and numerous clinical trials underway, the use of these drugs will only increase over time. More recent studies (such as Keynote-189, Keynote-407 and IMPOWER 150) showed that combining immunotherapy with other anticancer drugs such as chemotherapy and anti-angiogenics can lead to better response rates and survival outcomes [16,17]. Another area where immunotherapy is being investigated is earlier on in the treatment journey of a patient, such as the neoadjuvant setting, with a number of clinical trials underway. Given that more patients are likely to be exposed to immunotherapy over time, and with up to 15–25% experiencing potentially serious irAEs [1,3,4], we have to remain vigilant and well-prepared to manage these toxicities. 

Although immunotherapy has revolutionised the treatment of certain advanced cancers, the majority of patients will not derive any benefit [1,2]. Tumoural PD-L1 has the most robust body of evidence as a predictive biomarker and is the most widely available biomarker in clinical use. However, some patients treated with immunotherapy still showed an OS and PFS benefit even with a negative PD-L1 expression [17,34]. While TMB has been used in some clinical settings, it remains generally reserved for use in research trials. Thus, for the majority of patients commencing immunotherapy, there is lack of optimised and validated predictive biomarkers for clinical use. 

A number of potentially promising biomarkers have been discussed in this review from tissue, blood and the microbiome. Apart from tumoural PD-L1 and TMB which have already been discussed, novel STING and ASC are especially promising given the potential to manipulate these DNA damage pathways; however, more research in humans needs to be carried out in order to optimise their use. A disadvantage of tumour-based biomarkers, however, is that biopsies/surgery performed sometimes years prior to commencement of immunotherapy may not be representative of the current immunological status of the tumour. Blood-based “liquid biopsies” on the other hand can be repeated multiple times on treatment and have the advantage of being “less invasive”. However, the different blood-based biomarkers each require expertise and training in different methods of isolation and analysis; also, associated costs can be very high depending on specific techniques being used. Finally, the microbiome has potential to be manipulated to improve outcomes with seemingly simple methods such as using prebiotics, probiotics and faecal microbiome transplantation; however, the stigma of providing “stool” samples remains a barrier for its use in clinical practice.

It is unlikely that one biomarker will work for all patients and in all cancers. What is more likely is that different biomarkers will be found more useful in some patients than others and in some cancers more than others. For a biomarker to be used in routine clinical practice, a number of factors should be met, including having high sensitivity and specificity, having consistent results over time, being accessible and relatively affordable and finally having a quick turnaround to provide results to clinicians in a timely fashion. Potential biomarkers need to be evaluated in larger trials and the results validated. It can be a long process and can take many years to develop. The focus should always remain on personalising therapy to ensure the right person receives the right treatment. 

## 8. Conclusions

ICI therapy has become a mainstream treatment option for a number of advanced cancers over the past decade. However, lack of benefit in a significant proportion of patients, financial burden and potentially serious adverse events need to be considered when prescribing these drugs. Improving patient selection with the use of predictive/prognostic biomarkers is therefore an area of critical need. In addition to tumoural PD-L1 expression and TMB which are validated and in clinical use, several other potential biomarkers have/are being evaluated in the setting of advanced cancer treated with ICIs, including tumour-derived STING, ASC and TILs, blood-derived PBMCs, ctDNA, exosomes and cytokines, and finally, the microbiome. By continuing our research in this field, we hope to find that needle in the haystack. 

## Figures and Tables

**Table 1 cancers-13-00277-t001:** FDA approved ICIs.

Name	Target	Approved
**ipilimumab**	CTLA-4	2011
**nivolumab**	PD-1	2014
**pembrolizumab**	PD-1	2014
**atezolizumab**	PD-L1	2016
**avelumab**	PD-L1	2017
**durvalumab**	PD-L1	2017
**cemiplimab**	PD-1	2018

**Table 2 cancers-13-00277-t002:** Common irAEs.

irAE	Onset	Grade	Manifestation	Treatment
**Dermatitis**	Early (weeks)	1–2	Pruritis, maculopapular rash predominantly on trunk and extremities with facial sparing	Topical steroids or oral antihistamines, systemic steroids for Grade 3–4
**Enterocolitis**	Intermediate (weeks to months)	1–2, can be more severe	Diarrhoea, nausea, vomiting, abdominal pain, rectal bleeding	Antidiarrheal agents; Grade 3–4 is an emergency requiring supportive cares and systemic steroids
**Thyroid dysfunction (hypo or hyperthyroidism)**	Intermediate (weeks to months)	1–2	Signs/symptoms of hypo or hyperthyroidism	Hypothyroidism—thyroxine Hyperthyroidism—anti-thyroid drugs, radioactive iodine
**Hepatitis**	Intermediate (weeks to months)	1–2	Acute transaminitis	Systemic steroids in more severe cases

**Table 3 cancers-13-00277-t003:** Summary of ICI therapy in NSCLC, melanoma, RCC and urothelial/bladder cancer.

Cancer Type	ICI Used +/− Other Systemic Therapy	Indication	Evidence/Trial Name
**NSCLC**	pembrolizumab	1st line metastatic PD-L1 >50%	Keynote-024
	pembrolizumab + carboplatin + pemetrexed	1st line metastatic	Keynote-189, Keynote-407
	atezolizumab + carboplatin + paclitaxel + bevacizumab	1st line metastatic	IMPOWER 150
	nivolumab	2nd line metastatic	Checkmate-017, Checkmate-057
	pembrolizumab	2nd line metastatic	Keynote-010
	atezolizumab	2nd line metastatic	POPLAR
	durvalumab	Maintenance post definitive chemoRT locally advanced	PACIFIC
**Melanoma**	nivolumab	Adjuvant for resected stage III/IV	Checkmate-238
	pembrolizumab	Adjuvant for resected stage III	Keynote-054
	nivolumab + ipilimumab	Adjuvant for resected stage IV	IMMUNED
	pembrolizumab	Unresectable stage IV	Keynote-006
	nivolumab	Unresectable stage IV	Checkmate-066
	nivolumab + ipilimumab	Unresectable stage IV	Checkmate-067
**RCC**	nivolumab + ipilimumab	1st line metastatic	Checkmate-214
	nivolumab	2nd line metastatic	Checkmate-025
**Urothelial/bladder**	pembrolizumab	2nd line metastatic	Keynote-045
	avelumab	Maintenance post 1st line chemo metastatic	JAVELIN 100 bladder

**Table 4 cancers-13-00277-t004:** Summary of predictive capacity of tissue, blood and microbiome biomarkers.

Biomarker	Predictive Capacity
**Tumour Tissue**	
**PD-L1**	Increased expression associated with improved response to ICIs (>50% predicts response to pembrolizumab as 1st line therapy in NSCLC [1]
**TMB**	High TMB (>13Mut/Mb) associated with improved response to ICIs [35]
**STING**	STING deficient mice showed reduced therapeutic effect of ICIs, chemotherapy and radiotherapy [49]
**ASC**	ASC gene methylation predicted worse survival and increased tumour grade in mouse models [56]
**TME**	Higher density of CD3+, CD8+ and CD45RO+ T cells associated with improved survival and response to ICI therapy [31,61] Higher density of Tregs associated with worse survival and increased recurrence [31] TAMs associated with worse survival [63]
**Blood**	
**PBMCs**	High CD4+ and CD8+ T cells predicted improved response to ICI therapy in melanoma patients [66,67] ANC predicts poor survival [6,31] ALC predicts good survival [6,31] NLR >5 predicts worse survival [31,70,71,72,73,74] High LDH and CRP predict worse response [32] MDSCs associated with resistance to ICIs [8]
**ctDNA**	Reduction in ctDNA at 8, 12-week mark compared with baseline predicted response to ICI therapy in melanoma patients [79,80] Increase in ctDNA of >20% at 6-week mark compared with baseline predicted worse survival in NSCLC patients treated with ICIs [81]
**Exosomes**	Change in exosomal PD-L1 during ICI therapy in melanoma patients predicted response [6,31] Increase in exosome molecular cargo seen in non-responders in H and N cancer patients receiving cetuximab, ipilimumab and radiotherapy [88] Levels of exosomes from T cells and DCs increased in melanoma patients who responded to ipilimumab [89]
**Cytokines**	11 cytokines predicted irAEs in metastatic melanoma patients [92]
**Metal chelators**	Copper chelating drugs lead to increase in CD8+ T cells and NK cells and slowed tumour growth and improved survival in mice [91]
**sPD-L1 and sPD-1**	Time to treatment failure (TTF) and OS were significantly longer in those with low plasma sPD-L1 levels compared with high levels [97]
**CTCs**	Presence usually associated with poor response [99]
**Microbiome**	
	In melanoma patients who received ICIs, Faecalibacterium species enriched in responders while Bacteroides species enriched in non-responders [31] Diversity of gut microbiome associated with improved response to ICIs [6] Antibiotic use within 30 days of commencing ICI therapy associated with increased risk of early progression and reduced survival [105,106]

## Data Availability

Not applicable.

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
