# Peer review of "The Multiple Potential Biomarkers for Predicting Immunotherapy Response—Finding the Needle in the Haystack"

_cancers, 2021, doi:10.3390/cancers13020277_

Round 1
Reviewer 1 Report
Adam and collaborators presented a review entitled “The multiple potential biomarkers for predicting immunotherapy response – finding the needle in the haystack”, focusing on the description of some potential biomarkers for ICI response.
This review described the most relevant candidate biomarkers for ICI treatment, but, to date, only PD-L1 tumor level is validated, with some unsolved problems (i.e., patients with null PD-L1 that are responsive to ICI).
The manuscript is interesting and the topics are well described, but I suggest to implement this review with some information about soluble forms of PD-1 and PD-L1.
In particular, the authors give only a brief description of the soluble form of PD-L1 as potential biomarkers, but the literature presents, to date, some interesting data about the role of sPD-L1. I suggest inserting a more detailed description of the correlation between the response to ICI and the level of sPD-L1 in plasma patients. Moreover, I also suggest inserting in this review the potential role of the soluble form of the other member of this immune checkpoint, such as PD-1.
Minor points: I suggest a full revision of the punctuation
Lane 131: please remove S from options.
Lane 133: please add a comma in between refs. 91 and 92.
Lane 339: please remove the semicolon.
Lane 388: please remove the semicolon.
Lane 406; please remove the semicolon.
Author Response
Response to Reviewer 1 Comments
Point 1: The manuscript is interesting and the topics are well described, but I suggest to implement this review with some information about soluble forms of PD-1 and PD-L1. In particular, the authors give only a brief description of the soluble form of PD-L1 as potential biomarkers, but the literature presents, to date, some interesting data about the role of sPD-L1. I suggest inserting a more detailed description of the correlation between the response to ICI and the level of sPD-L1 in plasma patients. Moreover, I also suggest inserting in this review the potential role of the soluble form of the other member of this immune checkpoint, such as PD-1.
Response 1: This is a very interesting point. sPD-L1 and sPD-1 show some promising results as potential biomarkers in the literature. I have included a paragraph on soluble PD-L1 and PD-1 in the blood derived biomarkers chapter which addresses methods of detection and gives examples of its use as a biomarker in patients with NSCLC. Please see new section 5.7 lines 454-467.
Point 2: I suggest a full revision of the punctuation
Response 2: A full revision of the punctuation has been carried out.
Point 3: Lane 131: please remove S from options.
Response 3: S removed from options.
Point 4: Lane 133: please add a comma in between refs. 91 and 92.
Response 4: Comma added between refs. 91 and 92.
Point 5: Lane 339: please remove the semicolon.
Response 5: Semicolon removed.
Point 6: Lane 388: please remove the semicolon.
Response 6: Semicolon removed.
Point 7: Lane 406; please remove the semicolon.
Response 7: Semicolon removed.
Reviewer 2 Report
I enjoyed reading this review. However I was most interested in the cellular biomarkers which may predict (or not predict) response to specific monoclonal antibodies according to mechanism (for instance PD-L1 > 50% and response to pembrolizumab). If this data is missing from the literature, or there are negative reports, I would discuss this is the appropriate sections (see attached PDF).
Also, the sections may need to be reformatted so they give the correct subheading numbers.

Author Response
Response to Reviewer 2 Comments
Point 1: I enjoyed reading this review. However I was most interested in the cellular biomarkers which may predict (or not predict) response to specific monoclonal antibodies according to mechanism (for instance PD-L1 > 50% and response to pembrolizumab). If this data is missing from the literature, or there are negative reports, I would discuss this is the appropriate sections (see attached PDF).
Response 1: Thank you for your response. In the melanoma patients, all the studies referenced (86-92) evaluated PD-L1 expression in these patients. However, PD-L1 expression was not useful as patients benefited regardless of PD-L1 expression. In the renal cancer patients, Checkmate-025 did not evaluate PD-L1 expression and Checkmate-214 evaluated PD-L1 as an exploratory outcome; although patients with PD-L1 >1% had improved ORR and PFS, even patients with PD-L1 <1% benefited and OS was longer in the ICI group across all PD-L1 expression groups. I have discussed the above in the appropriate sections. Please see section 3.2 lines 135-137 and section 3.3 lines 147-50 and 153.
Point 2: Also, the sections may need to be reformatted so they give the correct subheading numbers.
Response 2: The sections have been reformatted to give the correct subheading numbers.
Reviewer 3 Report
Thank you for this great review on the burning question of how to solve the question of deciding on which patient to use ICIs or not.
This question is even more important since ICIs are slowly but surely moving into the first line treatment and still a good biomarker is missing.
minor typos: e.g. side 2: (CD)4+, I see an error concerning the numbering of chapters. each time a new chapter starts, it is 1. This should be addressed
Further those 2 markers STING and ASC are described more as new treatment modality, although the subject is focusing on biomarker. Are there results that underline its use as biomarker?
Talking about liquid biopsy maybe a word on CTCs (circulating tumor cells) should be included in the manuscript.
I like the summarized overview you give in table 4, although while reading this doesn't matches the text flow. maybe a few word could introduce the summary.
Besides from the described irAEs, there is also a need to better predict ICI treatment when thinking about pseudo-tumor-progress for example. The mentioned markers could also be used to better surveil cancer treatment.
Author Response
Response to Reviewer 3 Comments
Point 1: minor typos: e.g. side 2: (CD)4+, I see an error concerning the numbering of chapters. each time a new chapter starts, it is 1. This should be addressed
Response 1: The whole article has been reviewed again to fix the minor typos. The issue with numbering of chapters has been addressed, it now correctly reflects the appropriate chapter and subheadings.
Point 2: Further those 2 markers STING and ASC are described more as new treatment modality, although the subject is focusing on biomarker. Are there results that underline its use as biomarker?
Response 2: These 2 novel biomarkers do have a role as potential biomarkers.
Absence/deficiency of STING has demonstrated reduced efficacy to ICIs in mice models (ref 60) see lines 256-259. In STK11/LKB1 mutation associated STING deficiency predicted poor outcomes in NSCLC cell lines (ref 103) see lines 260-264. Although STING may be a potential therapeutic target, I still think it has a role as a biomarker as STING expression may predict patient response to ICIs.
Similarly, ASC methylation is associated with worse outcomes in NSCLC patients and correlated with high tumour grade in RCC cell lines (ref 78). This is still to be investigated in patient populations.
Point 3: Talking about liquid biopsy maybe a word on CTCs (circulating tumor cells) should be included in the manuscript.
Response 3: Detection of CTCs may reflect disease burden and their presence is usually associated with poor prognosis. I’ve included a paragraph about CTCs in the blood derived biomarkers chapter. Please see new section 5.8, lines 468-473.
Point 4: I like the summarized overview you give in table 4, although while reading this doesn't matches the text flow. maybe a few word could introduce the summary.
Response 4: Thank you for this point. I agree that an introduction to the summary table is required to provide better flow. I have addressed this, please see lines 518-519.
Point 5: Besides from the described irAEs, there is also a need to better predict ICI treatment when thinking about pseudo-tumor-progress for example. The mentioned markers could also be used to better surveil cancer treatment.
Response 5: I agree that we need to be better at predicting ICI treatment with regards to pseudo-progression. Although rare, it can cause significant distress to patients and can be clinically challenging to differentiate pseudo-progression from true progression. There may be a role for liquid biopsies (such as ctDNA) to differentiate between pseudoprogression and true progression. I have included relevant information in the ctDNA section (5.3, lines 397-402).